



# The influence of $^{14}CO_2$ releases from regional nuclear facilities at the Heidelberg $^{14}CO_2$ sampling site (1986 - 2014)

Matthias Kuderer, Samuel Hammer and Ingeborg Levin

Institut für Umweltphysik, Heidelberg University, Heidelberg, 69120, Germany

5    *Correspondence to*: Ingeborg Levin (Ingeborg.Levin@iup.uni-heidelberg.de)





**Abstract.** Atmospheric $\Delta^{14}CO_2$ measurements are a well established tool to estimate regional fossil fuel-derived $CO_2$ fluxes. However, emissions from nuclear facilities can significantly alter the regional $\Delta^{14}CO_2$ level. In order to accurately quantify the $^{14}CO_2$ signal, a correction term for anthropogenic sources has to be determined. In this study, the HYSPLIT atmospheric dispersion model has been applied to calculate the correction term for the long-term $^{14}CO_2$ monitoring site in Heidelberg. Wind

fields with a spatial resolution of 2.5° x 2.5°, 1° x 1° and 0.5° x 0.5° show systematic deviations, with coarser resolved wind fields leading to higher mean values for the correction. The mean correction for the period from 1986-2014, if based on the 0.5° x 0.5° wind field, which we assume as the most accurate, is 2.3 ‰ with a standard deviation of 2.1 ‰ and maximum values up to 15.2 ‰. After ceasing operations at the most important $^{14}CO_2$ source near Heidelberg in 2011, monthly nuclear correction terms decreased to less than 2 ‰, with a mean value of $(0.44 \pm 0.32)$ ‰ from 2012 to 2014.

## 1 Introduction

Evaluation of the perturbation of atmospheric $^{14}C$ by nuclear bomb tests in the middle of the last century has given very useful insight into carbon cycle dynamics (e.g. Levin and Hesshaimer, 2000). Today this artificial spike has almost equilibrated with the fast exchanging carbon reservoirs, and the currently observed global $\Delta^{14}CO_2$ trend is almost exclusively due to the ongoing input of $^{14}C$-free fossil fuel $CO_2$ into the atmosphere (Naegler and Levin, 2009; Levin et al., 2010; Graven, 2016). This long-

term trend can potentially be used to estimate the global input of fossil fuel $CO_2$ into the atmosphere. However, the uncertainty of this estimate is still large (ca. 30%, Levin et al., 2010) due to the uncertainty of the large $^{14}CO_2$ disequilibrium fluxes from biosphere and ocean, as well as artificial $^{14}C$ sources. On the continental scale, however, atmospheric $\Delta^{14}CO_2$ measurements provide a powerful and the only direct and quantitative tool for estimating the regional fossil fuel component. $^{14}CO_2$ measurements at a polluted station allow separating fossil fuel-derived regional $CO_2$ enhancements relative to a clean reference

level from those originating from biospheric fluxes if also the $^{14}CO_2$ level at the reference site is known (Levin et al., 2003; Turnbull et al., 2009). However, on that local to regional scale (several 10 km) $^{14}CO_2$ emissions from nuclear facilities, such as boiling water reactors, can significantly contaminate atmospheric $^{14}CO_2$. The $^{14}C$ signals from such point sources are well detectable in their immediate neighborhood in atmospheric $CO_2$ (and $CH_4$, e.g. Levin et al., 1992) but also in plant samples (Levin et al., 1988). $^{14}CO_2$ "plumes" from point sources normally quickly disperse at distances of some tens of kilometers. But

if a sampling station is located in the catchment of such $^{14}CO_2$ point sources, special care is required to accurately quantify the $^{14}CO_2$ contamination and correct for it to estimate reliable fossil fuel $CO_2$ values (e.g. Levin et al., 2003).

Here we present results from HYSPLIT dispersion modelling (Draxler and Hess, 1998) of $^{14}CO_2$ emissions from five nuclear installations in the < 60 km neighborhood of our long-term $^{14}CO_2$ monitoring site in Heidelberg. We apply the HYSPLIT

model for the period of 1986-2014 with available wind fields of 2.5° x 2.5°, 1° x 1° and 0.5° x 0.5° resolution. Using reported $^{14}CO_2$ emission rates, these model estimates for the Heidelberg sampling site allow us to correct for the local $^{14}CO_2$ contaminations from nuclear facilities (Kuderer, 2016). Our model results, however, turned out to strongly depend on the



resolution of the wind field used for the calculation. We discuss this important finding and present the currently most reliable corrections of our long-term $^{14}CO_2$ measurements.

## 2 Methods

### 2.1 Site description

The Heidelberg $^{14}CO_2$ sampling site is located on the University campus in the outskirts of Heidelberg, a medium size city in the upper Rhine valley in southwestern Germany (49° 25' N, 8°41'E, 116 m a.s.l., and see Figure 1). From 1986-2001, $^{14}CO_2$ samples have been collected from the roof of the former building of the Institute (INF 366) and from 2001 to present, from the new building about 500 m to the east (INF 229). At both locations, air was sampled from about 25 – 30 m a.g.l. The small difference in location of the two sampling sites is not relevant when estimating the nuclear $^{14}CO_2$ contamination with
HYSPLIT.

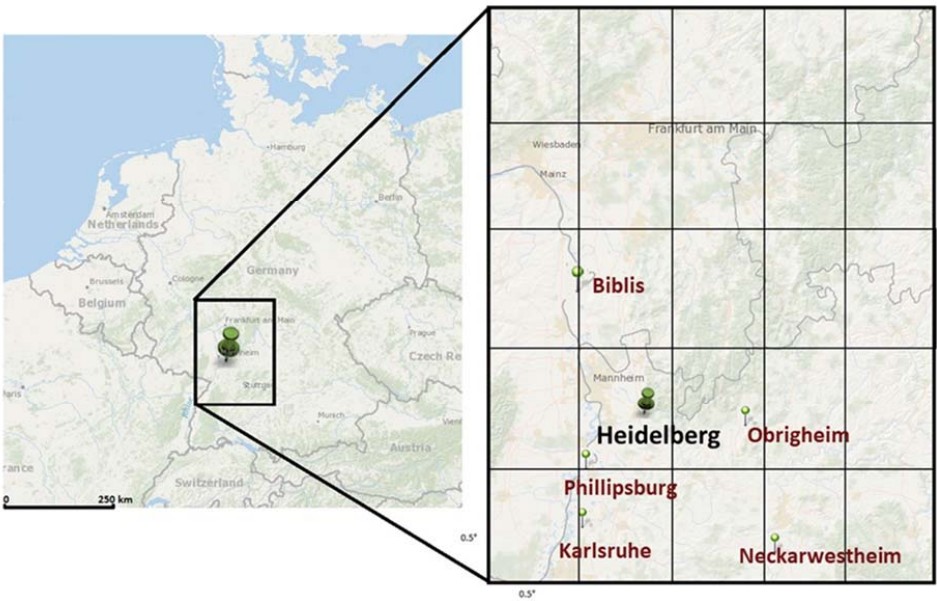

**Figure 1:** Map of the Heidelberg sampling site in southwest Germany. The locations of the five nearest nuclear facilities are shown in the
enlargement. This enlargement corresponds to the size of the 2.5° x 2.5° wind field grid. The 0.5° x 0.5° wind field resolution is indicated by the grid in the enlargement.



Five nuclear installations with reported $^{14}CO_2$ emissions are found at distances between 25 km and 55 km to the Heidelberg station. Figure 1 shows their locations; details of reactor type, installed electrical output, period of operation, distance from the

Heidelberg station and mean reported $^{14}CO_2$ emission during their operation up to 2014 are listed in Tab. 1. As the prevailing winds in the Upper Rhine valley are from south-west, Philippsburg (KKP I & II) is the most important source of potential $^{14}CO_2$ contamination in Heidelberg. Philippsburg I is the only boiling water reactor (BWR) with its major $^{14}C$ emissions being $^{14}CO_2$, whereas pressurized water reactors (PWR) emit $^{14}C$ mainly as $^{14}CH_4$. All the other nuclear installations except for Neckarwestheim II (GKN II) emit less than 15 % of Philippsburg I. Neckarwestheim is, however, located to the southeast of

Heidelberg in the Neckar valley at a distance of 55km, so that its relative contribution to the total $^{14}CO_2$ contamination is only less than 10 % (see Table 2).

### 2.2 $^{14}CO_2$ sampling and analysis

Two- and, for limited periods, also one-week integrated large volume samples of atmospheric $CO_2$ were collected from the roof of the Institute's buildings by quantitative chemical absorption in basic sodium hydroxide (NaOH) solution, as described

by Levin et al. (1980). Except for the first few years, samples were collected only during night (from 19:00 to 7:00 Central European Winter Time), in order to avoid $CO_2$ contamination from local traffic. Moving the Institute to a new building in the year 2000 required parallel $CO_2$ sampling at both, the old and the new sampling locations on the Heidelberg University campus, in order to quantify possible differences and then allow combining the data sets from the two locations about 500 m apart. As the new building is located closer to the Heidelberg city center, slightly lower $\Delta^{14}C$ values (by on average 0.8 ‰) were found

at the new location over the more than one-year overlapping period from late 2000 to early 2002. The results obtained from samples collected until 2002 at INF 366 at about 25 m a.g.l. were adjusted accordingly, and are now comparable with those obtained at the current sampling location at INF 229 at about 30 m a.g.l. (for details of this comparison and correction, see Levin et al. (2008)).

$^{14}CO_2$ samples were processed in the Heidelberg $^{14}C$ laboratory by acidification of the NaOH solution in a vacuum system. The extracted $CO_2$ was subsequently purified over charcoal. The $^{14}C/C$ ratio was then measured by low level counting (Kromer and Münnich, 1992). All results are presented here as $^{13}C$-corrected $\Delta^{14}C$ deviations from the international reference standard (Oxalic acid) in permil. They are corrected for decay to the date of $CO_2$ sampling (Stuiver and Polach, 1977). Note that Stuiver and Polach (1977) refer to this $^{14}C$ notation as $\Delta$ not $\Delta^{14}C$, however in order to be consistent with other atmospheric radiocarbon

literature we stick to using $\Delta^{14}C$ instead of $\Delta$. Precision of $\Delta^{14}C$ values was of order 4-5 ‰ in the 1980s and 1990s, of 3-4 ‰ in the 2000s and of 2-3 ‰ thereafter.



**2.3 Reported $^{14}CO_2$ emissions from nuclear facilities in the surroundings of Heidelberg**

According to the German Atomic Energy Act (Strahlenschutzverordnung, 2001), emissions of radioactive substances from nuclear facilities with the exhaust air must be monitored and reported quarterly to regional and federal authorities. The Bundesamt für Strahlenschutz (BfS, German Federal Office for Radiation Protection), releases yearly reports on radioactive

5 emissions from all German reactors and research facilities; here the $^{14}CO_2$ emissions are reported separately from other radioactive substances. These BfS reports are available for the years 1986 – 2014. For Philippsburg I and II higher resolution, i.e. monthly emission data are available (KKP pers. comm.); these monthly data were used in this work to estimate the $^{14}CO_2$ contamination in Heidelberg.

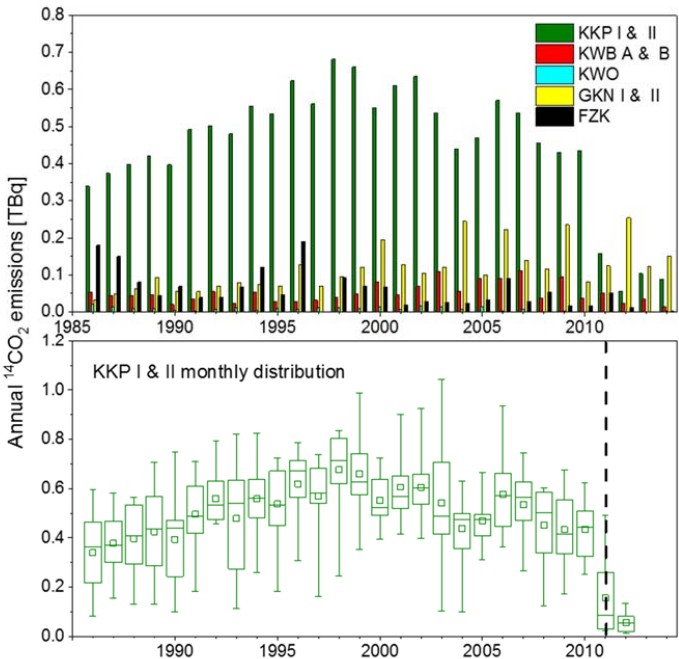

10 **Figure 2**: $^{14}CO_2$ emissions from nuclear facilities: Annual mean emissions from all facilities (upper panel) and box plots of the distribution of monthly values from Philippsburg KKP I & II (lower panel); the boxes include 50% of all months of the year with the horizontal bar indicating the mean and the square indicating the median value of the year. The whiskers show the minimum and maximum monthly values of the individual years. The dashed line indicates the shutdown of KKP I shortly after the Fukushima accident**.**




Figure 2 (upper panel) shows annual $^{14}CO_2$ emissions from 1986 – 2014 for all five facilities listed in Tab. 1, while Fig. 2 (lower panel) shows the distribution of monthly emissions from Philippsburg I and II for the years 1986 - 2012. Note the huge variability of monthly emissions, which can differ from month to month by more than a factor of two. Graven and Gruber

(2011) estimated mean emission factors of 0.06 TBq $^{14}CO_2$ GWa$^{-1}$ for PWRs and 0.51 TBq $^{14}CO_2$ GWa$^{-1}$ for BWRs. From our emission data and corresponding power production reports, we do see, however, large differences from these emission factors and for PWRs no correlation at all, as displayed in Fig. 3. Moreover, keeping in mind the huge month-to-month variability of $^{14}CO_2$ emissions from Philippsburg KKP I & II (Fig. 2, lower panel), underlines the necessity of reliable high-resolution $^{14}CO_2$ emission data from nuclear installations, if accurate corrections shall be applied to atmospheric $^{14}CO_2$ observations for fossil

fuel $CO_2$ estimates.

**Table 1:** Nuclear facilities in the surroundings of Heidelberg. Reactor type (BWR: boiling water reactor, PWR: pressurized water reactor), installed electrical power and the average annual $^{14}CO_2$ emissions during their respective period of operation up to 2014 as well as the
distance to the Heidelberg sampling site are given. Different reactor blocks are separated by slash. RR are research reactors and RP is the research reprocessing plant (WAK) of the Karlsruhe Research Center (FZK). After the operation period, further emissions occur during the decommissioning of the facilities.

| Nuclear facility | Installed electric capacity (MWe) | Type | Operation period | Mean $^{14}CO_2$ emission (TBq/yr) | Distance from Heidelberg |
|---|---|---|---|---|---|
| Philippsburg (KKP) I/II | 926/1468 | BWR /PWR | 1980-2011/ 1984-2019 | 0.414/0.055 | 25 km |
| Obrigheim (KWO) | 357 | PWR/PWR | 1969-2005 | 0.008 | 30 km |
| Biblis (KWB) A/B | 1225/1300 | PWR/PWR | 1975-2011 | 0.025/0.037 | 37 km |
| Karlsruhe FZK/WAK | - | RR/RP | 1971-1991 | 0.036 | 39 km |
| Neckarwestheim (GKN) I/II | 840/1400 | PWR/PWR | 1976-2011/ 1989-2022 | 0.008/0.135 | 55 km |



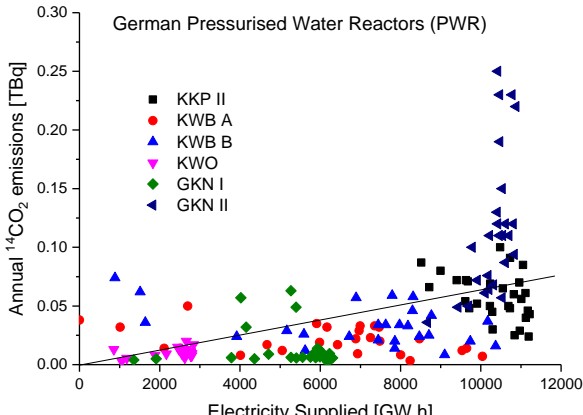

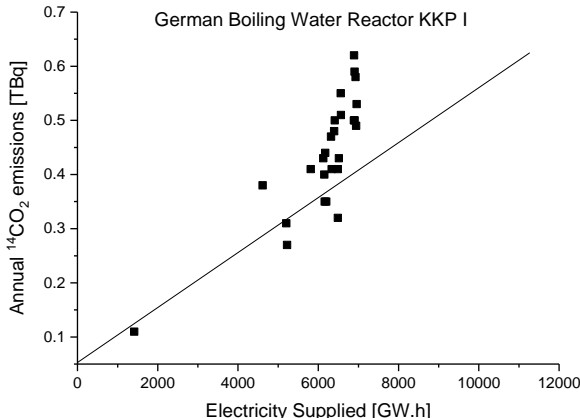

5   **Figure 3:** Relationship between annual $^{14}CO_2$ emissions from Pressurized Water Reactors (upper panel) and the Boiling Water Reactor Philippsburg I (lower panel) and their annual electricity supplied. The solid lines show the specific emission factors reported by Graven and Gruber (2011).



### 2.4 The HYSPLIT model

The Hybrid Single-Particle Lagrangian Integrated Trajectory model (HYSPLIT) from NOAA offers a variety of services ranging from computing simple air parcel trajectories up to complex dispersion simulations (Draxler and Hess 1998). During the simulations, virtual particles are emitted at the source location and advected to the new particle position, described by the position vector **P**, using the input wind velocity vector field **V**:

$$\mathbf{P}(t + \Delta t)\_advection = \mathbf{P}(t) + 0.5 \cdot [\mathbf{V}(P, t) + \mathbf{V}(\mathbf{P'}(t+\Delta t), t+\Delta t)] \cdot \Delta t. \tag{1}$$

The advection equation is solved with a dynamic time step $\Delta t$, demanding that the advective displacement is smaller than the size of a grid cell (Draxler, 1999). Equation 1 is solved numerically by integrating the velocity vector over time, making use of the trapezoidal rule, i.e. averaging the velocity vectors at the initial position $\mathbf{V}(\mathbf{P}, t)$ and first-guess position $\mathbf{V}(\mathbf{P'}(t+\Delta t), t+\Delta t) = \mathbf{V} \{(\mathbf{P}(t) + \mathbf{V}(\mathbf{P}, t) \cdot \Delta t), (t + \Delta t)\}$ of the particle. To account for atmospheric dispersion, the particles are displaced stochastically (Eq. 2a & b):

$$X\_final(t + \Delta t) = X(t + \Delta t)\_advection + U'\_dispersion(t + \Delta t) \cdot \Delta t \tag{2a}$$

$$Y\_final(t + \Delta t) = Y(t + \Delta t)\_advection + W'\_dispersion(t + \Delta t) \cdot \Delta t \tag{2b}$$

where the turbulent velocity components U', W' are estimated from the standard deviations $\sigma$ of the horizontal or respective vertical velocity components (Fay et al., 1995). For more details, see Stein et al. (2015) and references therein. After the advective and dispersive displacement, the HYSPLIT model computes the particle concentration in every grid cell, which gives a dilution factor f (see Eq. 3), describing how much the point source emissions are diluted over the respective grid. This dilution factor is strongly depending on the prevailing meteorological conditions.

### 2.5 Wind fields

Previous studies have shown that HYSPLIT calculations are sensitive to the meteorological input data (e.g., Cabello et al., 2008; Lin et al., 2015). Here we used three different wind velocity fields that have a horizontal resolution of 2.5° x 2.5°, 1° x 1° and 0.5° x 0.5°. The GDAS (Global Data Assimilation System) assimilates meteorological observations in numerical weather prediction models and archives the results. The one degree fields GDAS1 are available since 2005 and the half degree fields GDAS0p5 since 2008. GDAS1 and GDAS0p5 differ besides the horizontal also in the vertical resolution (Lin et al., 2015). The NCEP/NCAR (National Centre for Environmental Prediction/National Centre for Atmospheric Research) reanalysis provides atmospheric analyses with a spatial resolution of 2.5° x 2.5°, using historical data from 1948 onwards. All three wind fields are readily available at ftp://arlftp.arlhq.noaa.gov/pub/archives/.





### 2.6 Estimation of $\Delta^{14}C_{nuclear}$

The $^{14}C$ signal at the sampling site $\Delta^{14}C_{nuclear}$ originating from $^{14}CO_2$ emissions from each nuclear facility is calculated by scaling the meteorological dilution factor f (s m$^{-3}$) at the measurement station obtained from the HYSPLIT simulation with the time-varying emission strength Q (Bq s$^{-1}$) of the source. This specific $^{14}C$ activity is converted (according to its definition from

5   Stuiver and Polach (1977)) into $\Delta^{14}C_{nuclear}$ in ‰ according to Eq. 3

$$\Delta^{14}C_{nuclear} \ (‰) = f \cdot Q \cdot X_{CO2} / (M_C \cdot V_m \cdot a), \tag{3}$$

with the molar volume at standard atmospheric temperature and pressure (STP) $V_m = 24.465$ mole m$^{-3}$, molar mass of carbon $M_C = 12$ g mole$^{-1}$, mole fraction of $CO_2$, $X_{CO2}$, and specific activity of the $^{14}C$ standard a = 0.238 Bq gC$^{-1}$.

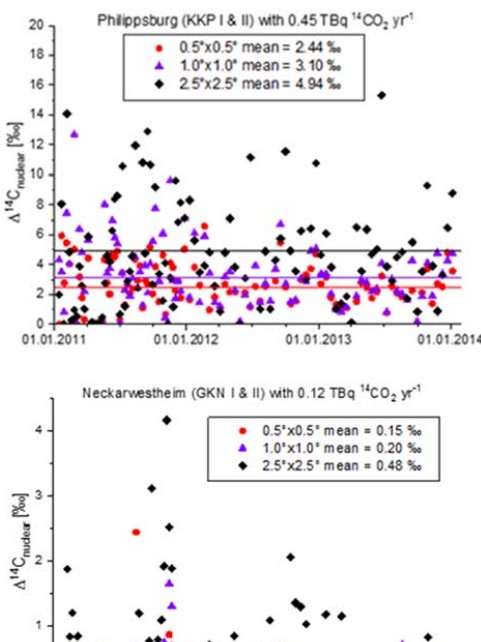

**Figure 4:** Upper panel: Calculated $\Delta^{14}C_{nuclear}$ contributions from Philippsburg (KPP I & II) with assumed constant $^{14}C$ emissions using the three wind fields with different resolution. Lower panel: Same as upper panel, showing the contributions from Neckarwestheim (GKN I & II).





# 3 Results

## 3.1 $\Delta^{14}C_{nuclear}$ estimates using wind fields of different resolution

Figure 4 (upper panel) shows two-weekly (i.e. sampling period) integrated HYSPLIT-estimated $\Delta^{14}C_{nuclear}$ contributions in Heidelberg for 2011 – 2013, originating from assumed constant $^{14}CO_2$ emissions from Philippsburg of 0.45 TBq yr$^{-1}$

(corresponding to the long-term average emission from this facility). The different symbols distinguish the results when using the three different wind fields, i.e. with resolution of 2.5° x 2.5° (black diamonds), of 1° x 1° (blue triangles) and of the highest resolution of 0.5° x 0.5° (red circles). The two-week integrated $\Delta^{14}C_{nuclear}$ signals vary between 0‰ and 16 ‰ for the coarse resolution wind field, and show on average lower signals when using the higher resolved wind fields. There are, however, also situations when we obtain lower contamination signals with the coarse resolution wind field than with the higher resolved

fields. The 1° x 1° wind field also yields, on average, slightly higher $\Delta^{14}C_{nuclear}$ signals from Philippsburg than the highest resolution 0.5° x 0.5° wind field, but the differences between those two are often only marginal. Looking at the contributions from the Neckarwestheim reactors (GKN I & II) (Figure 4 lower panel), we also estimate the largest $\Delta^{14}C_{nuclear}$ signals with the low-resolution wind field, while the highest resolution wind field yields the smallest signals. The mean ratio between the contamination signals estimated with the highest resolution wind field and those estimated with the 2.5° x 2.5° resolution field

is 0.43. We consider the results from the higher-resolution wind fields more reliable to calculate $\Delta^{14}C_{nuclear}$ than those with the coarse resolution field (see discussion below). We further conclude that the contributions from Neckarwestheim $^{14}CO_2$ emissions on the Heidelberg $\Delta^{14}CO_2$ signal are, on average, about one order of magnitude smaller than those from Philippsburg and, thus, with an average $\Delta^{14}C_{nuclear}$ of less than 0.2 ‰, almost negligible.

## 3.2 Estimation of $\Delta^{14}C_{nuclear}$ in Heidelberg from all five nuclear installations

Owing to its source strength and proximity to Heidelberg, Philippsburg is the dominant contributor to the nuclear contamination at our sampling site. Therefore, and considering the high month-to-month variability of emissions (Fig. 2, lower panel), it is important to use monthly resolved emission data to estimate the $\Delta^{14}C_{nuclear}$ signals originating from KKP I & II. The other four nuclear installations are secondary contributors permitting the use of annual average $^{14}CO_2$ emission rates in absence of higher temporally resolved emission data. For each source location, the HYSPLIT model was run for every calendar

day separately covering the period 1986 - 2014. Due to the small distance between $^{14}C$ sources and the measurement station, simulations were limited to 48h, where each run consisted of a 24-hour period, when particles were emitted with a constant rate, followed by 24 hours of sole propagation of the particles. Thus, for each day the simulated nuclear $^{14}C$ activity included the actual emissions of this day arriving at the sampling site and the propagated emissions from the day before. This could potentially lead to loss of particles, which arrive at the measurement site more than 24-48 hours after the release, but for an

extended reference period no such effect has been observed. Typical travel times from the nuclear power plants to Heidelberg are in the order of 6-12 hours.



**Table 2:** Relative average $\Delta^{14}C_{nuclear}$ contribution in Heidelberg from 1986 to spring 2011 (shutdown of KKP I)

|   | KWO | KWB A & B | GKN I & II | KKP I & II | FZK/WAK |
|---|---|---|---|---|---|
| % | 1.05 | 1.39 | 6.80 | 88.13 | 2.63 |

For the KKP reactor site, the following meteorological data has been used: For 1986 – 2008 and 2010, we used the 2.5° x 2.5° fields, for 2009 and 2011 – 2014 the 0.5° x 0.5° fields. For the other four source locations (KWO, KWB A & B, GKN 1 & 2 and FZK, WAK), the 2.5° x 2.5° wind field data have been used for the entire period 1986 – 2014, in order to save computing time. All coarse grid dilution factors were then corrected with a factor of 0.43 to account for the effect of under-estimating atmospheric dispersion in coarse grid simulations. This factor was obtained from the comparison made for the 3-year period

2011-2013 at Philippsburg and Neckarwestheim (Fig. 4). The average relative contributions to the total $\Delta^{14}C_{nuclear}$ signal for all facilities are listed in Tab. 2. The largest correction terms for a two-week sampling period originating from KKP I & II were 15.2 ‰, from GKN I & II, it was 3.3 ‰ and from KWB A & B it was 1.1 ‰. From the other two facilities, they were always smaller than 1 ‰.

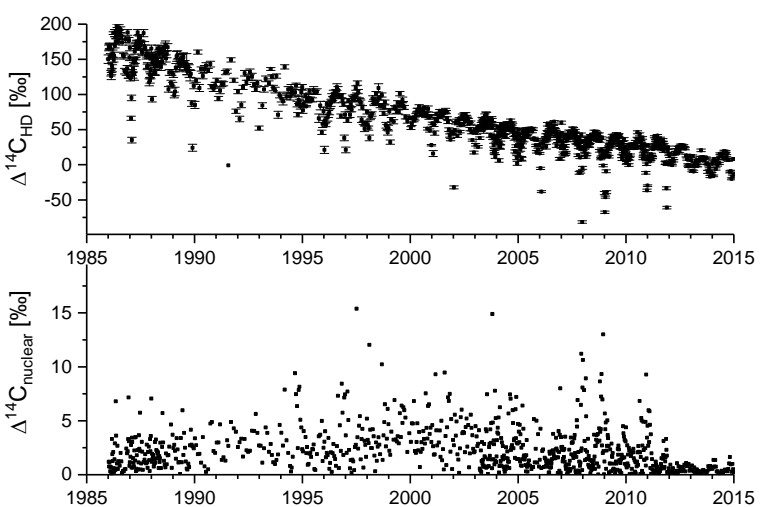

**Figure 5:** Upper panel: Results of $\Delta^{14}CO_2$ measurements in Heidelberg (uncorrected); lower panel: nuclear contribution from all installations in Heidelberg (note expanded $\Delta^{14}C$ scale)





The individual uncorrected $\Delta^{14}CO_2$ Heidelberg data are displayed in Fig. 5 (upper panel) together with the individual total

$\Delta^{14}C_{nuclear}$ corrections (lower panel). In the years before the KKP I shutdown, about 1 % of all corrections were above 10 ‰ and less than 2 % above 5 ‰. The mean correction was 2.3 ‰ with a standard deviation of 2.1 ‰. After the shutdown of the BWR KKP I, the largest $^{14}CO_2$ source before 2011, $\Delta^{14}C_{nuclear}$ decreased to less than 2 ‰, with a mean value of $(0.44 \pm 0.32)$ ‰ from 2012 to 2014. It is therefore feasible to apply only an average correction of this size to the Heidelberg measurements of all subsequent years.

**3.3 Uncertainty of estimated $\Delta^{14}C_{nuclear}$**

The uncertainty of our $\Delta^{14}C_{nuclear}$ estimates originates from uncertainties in emission data and uncertainties in the HYSPLIT model transport. From comparison of results based on the differently resolved wind fields (Fig. 4), we find the largest deviations between the 2.5° x 2.5° and the 1° x 1° fields while the average differences between the two finer resolved wind fields are of order 30 %, they can, however, be as large as a factor of two for individual two-week periods. The uncertainty of

the measured monthly emission data is probably less than 10-20 % and thus small if compared to the uncertainty of the model transport (although sub-monthly variability in the emissions may also contribute to the uncertainty of the $\Delta^{14}C_{nuclear}$ estimates). For the contributions from nuclear installations where only annual average emission data were available to us, the uncertainty of emissions is estimated to 30 %. As the contribution from all four installations except Philippsburg contribute on average only 12 % (Tab. 2) this uncertainty is small compared to the transport uncertainty of the contributions from Philippsburg. We,

therefore, estimate the typical uncertainty of individual total $\Delta^{14}C_{nuclear}$ signals to less than 35 %.

**4 Discussion and Conclusions**

Our HYSPLIT estimates of $^{14}CO_2$ contaminations from nuclear facilities in the catchment area of Heidelberg showed large differences when using wind fields of different resolution. The calculated mean contamination was approximately twice as large when using the coarse resolution 2.5° x 2.5° wind field compared to the two higher resolution fields. Previous studies

have shown, that meteorological coarse grid re-analyses can be well suited to capture synoptic-scale dynamical processes, but biases in surface wind speeds may be introduced as re-analysis data are not well adapted to reproduce transient strong wind events occurring at the mesoscale and generating a large sub-grid scale variability (Largeron et al., 2015). These can arise in HYSPLIT trajectory calculations, which are the basis for concentration simulations, when the air mass passes through areas with complicated topography and meteorological patterns that are on a smaller scale than the data resolution (Su et al., 2015).

Another and possibly more important factor is that atmospheric dispersion is included in the model by using the standard deviation of the interpolated velocity field. Linearly interpolating the coarse wind field to the internal HYSPLIT grid (here



0.05° x 0.05°) leads to a less variable velocity field compared to initially starting with a fine grid. This generates more distinct plume shapes in coarse grid simulations (Kuderer, 2016). Therefore, using the coarse wind field may underestimate the effect of atmospheric dispersion, leading to high values when the plume directly passes the measurement point. We expect this to occur frequently in the case of the Philippsburg $^{14}CO_2$ plume, where the source lies in the main wind direction at rather short

distance from the measurement point. In the case of Neckarwestheim, this explanation does not hold. However, also here we consider the results obtained with the finest resolution wind field as more accurate. GKN lies in the hilly Neckar valley with a complex topography, which is probably better represented by the finer resolution wind fields. Overall, we expect the HYSPLIT estimates that are based on higher resolution wind fields to provide more realistic results, in particular as the topography around Heidelberg is not flat. We therefore correct the HYSPLIT results obtained with the 2.5° x 2.5° wind fields for the earlier years

when high-resolution wind fields (0.5° x 0.5°) are not available (see above).

In an earlier study by Levin et al. (2003), KKP I & II were considered as the sole sources for the nuclear contamination at the Heidelberg sampling site. A Gaussian plume model (Turner, 1970) with a constant mean dispersion factor had been applied there to calculate $\Delta^{14}C_{nuclear}$ as a first approximation, but using the same monthly $^{14}CO_2$ emissions as in the present study. The

mean nuclear signal estimated by Levin et al. (2003) was $\Delta^{14}C_{nuclear} = (4.8 \pm 2.0)$ ‰ ranging from 0.2 ‰ to 10 ‰ for monthly mean values. This earlier estimate of $^{14}CO_2$ contamination is approximately twice the value obtained with the HYSPLIT model and the high-resolution wind fields. Graven and Gruber (2011) used the TM3 model with a spatial resolution of 1.8° x 1.8° and estimated for 2005 a total $\Delta^{14}C_{nuclear}$ of 2.1 (1.1 - 3.7) ‰ for the Heidelberg grid cell. Their estimate is in agreement with our results for that year ((2.1 ± 1.6) ‰) obtained with the high-resolution wind field. As in the present study, Graven and

Gruber (2011) also included $^{14}C$ contributions from other nuclear installations in their estimates. However, their assumed emissions from the Philippsburg I reactor were estimated with the average emission factor for BWR, which is about 20 % smaller than the measured value for 2005 used in our estimate. They also mention that their Eulerian model may have under-estimated the true contamination due to its coarse resolution, which would dilute point source emissions over a large grid in an Eularian approach.


These comparisons with earlier studies indicate that more work and higher resolution models and wind fields are needed to reduce the uncertainty of the $^{14}CO_2$ contamination estimates from nuclear installations at measurement sites where $\Delta^{14}CO_2$ observations shall be used to precisely determine the regional fossil fuel $CO_2$ component. Currently, we have to take into account a model transport uncertainty of about 1-2 ‰ in the estimated $\Delta^{14}C_{nuclear}$ contamination, if the measurement site is

located closer than about 30 km downwind from a nuclear facility, which has a $^{14}CO_2$ emission rate of about 0.5 TBq yr$^{-1}$ similar to the Philippsburg I boiling water reactor with 1 MWe power production. Other reactor types, such as the Canadian CANDU reactors may have significantly larger emission rates (Graven and Gruber, 2011; Vogel et al., 2013); the uncertainty of corresponding $\Delta^{14}C_{nuclear}$ estimates in their close neighborhood may then be considerably larger.




The limited accuracy and temporal resolution of $^{14}CO_2$ emission rates from nuclear installations cause additional uncertainty on the $\Delta^{14}C_{nuclear}$ estimates, as generally only annual mean emissions are reported. Graven and Gruber (2011) assume that $^{14}CO_2$ emissions are proportional to the annual power production. However, the present study on the influence from German

reactors on the Heidelberg measurement site does not fully support this finding. Figure 4 does not show significant correlations between annual $^{14}CO_2$ emissions and corresponding electricity supply. Therefore, assuming emission factors as suggested by Graven and Gruber (2011) will add considerable uncertainty to the $\Delta^{14}C_{nuclear}$ estimates, which may be as large as the uncertainties estimated here for model transport error.

Overall, we conclude that careful investigation of potential $^{14}CO_2$ emissions in the catchment of sampling sites is required when using $^{14}CO_2$ observations for fossil fuel $CO_2$ estimates. The differences of our modelling results, when based on differently resolved wind fields, together with the findings from earlier studies suggest that current $\Delta^{14}C_{nuclear}$ estimates may be wrong by a factor of two. Therefore, careful investigations with high-resolution models must be performed at all stations where $^{14}C$-based fossil fuel $CO_2$ measurements are conducted. We plan such studies for the European ICOS atmospheric station

network (https://www.icos-ri.eu/icos-stations-network). The basis must be high-resolution $^{14}CO_2$ emissions data from nuclear facilities, which need to be made available for these investigations, if contamination estimates shall be accurate.

**Data availability**

Data will be available from the Heidelberg University data depository under
https://heidata.uni-heidelberg.de/dataverse/carbon

**Author contributions**

I.L. and S.H. have designed the study. M.K. made the HYSPLIT model calculations and evaluated the data. I.L. prepared the manuscript with support from M.K. and S.H.

**Acknowledgements**

The authors gratefully acknowledge the NOAA Air Resources Laboratory (ARL) for providing the HYSPLIT transport and
dispersion model used in this publication. We especially like to thank Bernd Kromer and the staff of the Heidelberg $^{14}C$ laboratory in Heidelberg for their careful work analysing the $^{14}CO_2$ samples, and Ute Karstens for helpful discussions on the manuscript. Financial support came from a number of agencies in Germany and Europe. These are the Heidelberg Academy of Sciences, the Ministry of Education and Science, Baden-Württemberg, Germany, the German Science Foundation, the



German Ministries for the Environment, of Education and Science, and of Transportation and digital Infrastructure, the German Umweltbundesamt and the European Commission, Brussels.

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
