# Peer review of "The influence of $^{14}\text{CO}_2$ releases from regional nuclear facilities at the Heidelberg $^{14}\text{CO}_2$ sampling site (1986 - 2014)"

_Atmospheric Chemistry and Physics, 2018_

## Referee Comment (RC1) · J. Turnbull (Referee) · 20 Feb 2018

This paper describes a new modelling study that evaluates the influence of local (<100 km distant) nuclear power plant 14C emissions on 14CO2 measurements at Heidelberg, Germany. They transport detailed reported emissions from the nearby power plants using the HySPLIT model at several different meteorology resolutions. They identify which power plants contribute significantly to 14CO2 at Heidelberg, and how that varies through time. The results show that higher resolution meteorological fields are helpful in evaluating the influence of point source emissions such as these. More importantly, they show that when looking at individual sites with nearby nuclear 14C

sources, annual or monthly emission data may be insufficient.

This paper has a well-defined topic that is clearly explained, it is well-written, and the results are clear. It is a nice contribution to the literature and will be particularly relevant to the atmospheric 14C community. I have only a few extremely minor comments to clarify particular points, and recommend that this paper be accepted with these very minor changes.

Specific comments: Pg 2 line 3 and Pg 3 line 1-2. You say "in order to quantify the 14CO2 signal", but I think you mean to say "in order to quantify the fossil fuel CO2 signal". The 14CO2 signal naturally includes all sources including nuclear contributions, it is the fossil fuel CO2 calculation that needs to be adjusted to account for nuclear emissions. Pg 2 line 14. Naegler and Levin 2009 and Graven 2016 are not in the reference list. Please check referencing throughout. Also, please use hanging indents or numbering for the reference list to make it easier to scan through. Pg 2 line 22. I am not sure that "contaminate" is the right word, "influence" would be better. Pg 2 line 22-24. There are a number of studies that have looked at 14C emissions from nuclear power plants, please reference some from research groups other than your own. For example: Povinec, P.P., Chudá, M., Šivo, A., Šimon, J., Holá, K., Richtáriková, M. Forty years of atmospheric radiocarbon monitoring around Bohunice nuclear power plant, Slovakia(2009) Journal of Environmental Radioactivity, 100 (2), pp. 125-130. Dias, C.M., Santos, R.V., Stenström, K., Nícoli, I.G., Skog, G., da Silveira Corrêa, R.14C content in vegetation in the vicinities of Brazilian nuclear power reactors(2008) Journal of Environmental Radioactivity, 99 (7), pp. 1095-1101. Koarashi, J., Akiyama, K., Asano, T., Kobayashi, H. Chemical composition of 14C in airborne release from the Tokai reprocessing plant, Japan (2005) Radiation Protection Dosimetry, 114 (4), pp. 551-555. Stenström, K., Erlandsson, B., Hellborg, R., Wiebert, A., Skog, S., Vesanen, R., Alpsten, M., Bjurman, B. A one-year study of the total air-borne14C effluents from two Swedish light-water reactors, one boiling water- and one pressurized water reactor(1995) Journal of Radioanalytical and Nuclear Chemistry Articles, 198 (1),

pp. 203-213. Uchrin, G., Hertelendi, E., Volent, G., Slavik, O., Morávek, J., Kobal, I., Vokal, B. 14C measurements at PWR-type nuclear power plants in three middle European countries(1998) Radiocarbon, 40 (1), pp. 439-446. Pg 4 line 5-8. Please include references to back the statement that BWR reactors mostly emit 14CO2 whereas others emit 14CH4. Table 1. Please include references for the information shown in the table. Pg 8 line 21-28. Are there previous studies that examined the performance of HySPLIT with met data at different resolutions? What did they conclude? Pg 11 line 5 and throughout. Through most of the paper, the nuclear facilities are identified by their names – "Phillipsburg", etc. Here they are identified by the 3 letter codes, which are particularly confusing since KPP is not obviously the same place as Phillipsburg. Choose either the names or 3 letter codes and stick with them throughout the text. Pg 14 lines 1-10. I agree that the detailed emissions and LaGrangian model used in this paper give more detail (and more variability) than Graven and Gruber showed in their earlier paper. Yet a little more nuance in this paragraph would be helpful. In cases where nuclear facilities are nearby and have a strong influence, the detailed studies such as this one will be necessary. But for continental-scale studies looking at monthly or annual resolution, the gridded datasets provided by Graven and Gruber will likely be sufficient – and in many cases, it may be difficult to get more detailed information, so the Graven and Gruber dataset may still be the best choice.

––––––––––––––––––––

---

## Referee Comment (RC2) · Anonymous Referee #2 · 2 Mar 2018

Kuderer et al. present an analysis of nuclear power plant influences on radiocarbon measurements in CO2 at Heidelberg using emissions data and the Hysplit model at three resolutions. Their main conclusions are that the nuclear correction decreased after the shutdown of Philippsburg BWR, the corrections they estimate are sensitive to model resolution, and nuclear corrections require careful consideration.

The authors' work is useful and important to the community. However, some revisions are needed to clarify the details of their study and to expand the conclusions drawn from their results.

The methods for model simulations are not very clear and there appear to be several

different simulations used that are rather hard to follow. A table describing the different simulations run for each nuclear site would be helpful. Details about how the Hysplit runs were conducted, such as the number of particles and release times should be added. The authors should also clarify that Hysplit was run in forward mode from the locations of the nuclear sites rather in backward mode from the observation site in Heidelberg. There appears to be some details described in the results section 3.2 that would fit better in the methods section.

The authors report in the abstract that "The mean correction for the period from 1986-2014, if based on the 0.5° x 0.5° wind field, which we assume as the most accurate, is 2.3 ‰[']. However, it appears high resolution 0.5° winds were only used in simulations for 2009 and 2011 – 2014, so it is not correct to say the 1986-2014 correction is from the 0.5° x 0.5° wind field. The other years were estimated from the coarser 2.5° resolution simulations with a correction factor based on comparisons for the years where 2.5° and 0.5° simulations were run for two reactors.

Since Fig 4 shows the difference between simulated corrections at different resolution for individual samples is sometimes very small and sometimes very large (even with fixed emissions), is it valid to apply a mean correction to the data before 2009? Particularly if a main argument the authors are making is that the correction is highly variable in time? The authors argue that, since the correction is highly variable in time, monthly emissions data must be used and average emissions cannot be used, but then seem to contradict themselves by saying an average correction can be applied to account for model resolution, when actually this can be highly variable as well. Another point is that 0.5° is still rather coarse compared to some regional modelling currently being done at 0.1° or finer resolution.

Why do the authors use fixed emissions in the simulations shown in Fig 4?

If the authors have simulations with both fixed and monthly-varying emissions, can they include a comparison of these two simulations to quantify variability due to vary-

ing emissions vs variability due to varying transport? This comparison would be very useful.

The authors note the previous estimate of the average nuclear correction by Levin using the plume model is higher than their estimate. Although the plume model is simpler, it might be considered to be at finer resolution than $0.5°$, and therefore a better estimate.

Can the authors make any inference on the detectability of the Philippsburg shutdown based on the Heidelberg Delta14CO2 data?

The authors should discuss the impact the Philippsburg shutdown would have on the inferred fossil fuel CO2 at Heidelberg, if the change in the nuclear correction after 2011 was not accounted for. How does the change in the nuclear correction compare to the average fossil fuel signal in Delta14CO2 at Heidelberg?

Section 3.3 Uncertainty in estimated nuclear correction – this needs more detail and seems rather too qualitative. The authors do not seem to include model transport uncertainty also for the high-resolution case. Do the authors have an estimate for the magnitude of sub-monthly variation in emissions?

Do the authors think that monthly resolution in emissions data is sufficient, in general? Would this depend on the sampling integration time?

Could Fig 2b show the time series of emissions rather than yearly boxplots? It would be interesting to know if there is any pattern to the emissions over the year – for example, are emissions typically higher in summer potentially related to more maintenance undertaken in summer?

The authors should consider the Cattenom reactor in France, to the west and upwind of Heidelberg. The authors should also consider if Heidelberg could sometimes be sensitive to emissions from La Hague, which is further away, but emits >20x more 14CO2 than Philippsburg.

Shouldn't Eq. 3 have a factor of 1000 for per mil units? What is used for XCO2 in this

calculation?

First sentence in abstract – last word "fluxes" should be deleted. Radiocarbon measurements quantify fossil fuel derived $CO_2$, but not fluxes. Also here the Delta notation is used without describing it. The phrase "$14CO_2$ signal" is unclear – do you mean nuclear $Delta14CO_2$ signal? Why are $Delta14CO_2$ and $14CO_2$ both used in the abstract? Isn't the Heidelberg site monitoring $Delta14CO_2$ rather than $14CO_2$? Last sentence should be revised to "After operations at the Philippsburg boiling water reactor ceased in 2011, the" . . .

P2, L22 Delete "well". L24 Comment about "normally quickly disperse" needs reference or should be deleted.

P10, L21-24 – Please show some quantitative evidence from the simulations to support these statements.

---

## Author Comment (AC1) · 30 Apr 2018

Replies to Referee #1, Jocelyn Turnbull

We wish to thank Jocelyn Turnbull for her comments and suggestions for changes; we have revised the manuscript as follows (our answers are given in blue in the text below)

This paper describes a new modelling study that evaluates the influence of local (<100 km distant) nuclear power plant 14C emissions on 14CO2 measurements at Heidelberg, Germany. They transport detailed reported emissions from the nearby power plants using the HySPLIT model at several different meteorology resolutions. They identify which power plants contribute significantly to 14CO2 at Heidelberg, and how that varies through time. The results show that higher resolution meteorological fields are helpful in evaluating the influence of point source emissions such as these. More importantly, they show that when looking at individual sites with nearby nuclear 14CThis paper has a well-defined topic that is clearly explained, it is well-written, and the results are clear. It is a nice contribution to the literature and will be particularly relevant to the atmospheric 14C community. I have only a few extremely minor comments to clarify particular points, and recommend that this paper be accepted with these very minor changes.

Specific comments: Pg 2 line 3 and Pg 3 line 1-2. You say "in order to quantify the 14CO2 signal", but I think you mean to say "in order to quantify the fossil fuel CO2 signal". The 14CO2 signal naturally includes all sources including nuclear contributions, it is the fossil fuel CO2 calculation that needs to be adjusted to account for nuclear emissions.

This is absolutely correct, we have changed the wording correspondingly.

Pg 2 line 14. Naegler and Levin 2009 and Graven 2016 are not in the reference list. Please check referencing throughout. Also, please use hanging indents or numbering for the reference list to make it easier to scan through.

Thank you for pointing this out. We have added the references. Concerning formatting, it is not our choice but the Copernicus word template, which asks for this formatting, which I also find very unpractical …

Pg 2 line 22. I am not sure that "contaminate" is the right word, "influence" would be better.

From our point of view it is a "contamination", and we would like to keep this expression, as "influence" is very unspecific.

Pg 2 line 22-24. There are a number of studies that have looked at 14C emissions from nuclear power plants, please reference some from research groups other than your own. For example:

Povinec, P.P., Chudá, M., Šivo, A., Šimon, J., Holá, K., Richtáriková, M. Forty years of atmospheric radiocarbon monitoring around Bohunice nuclear power plant, Slovakia (2009) Journal of Environmental Radioactivity, 100 (2), pp. 125-130.

Dias, C.M., Santos, R.V., Stenström, K., Nícoli, I.G., Skog, G., da Silveira Corrêa, R. 14C content in vegetation in the vicinities of Brazilian nuclear power reactors (2008) Journal of Environmental Radioactivity, 99 (7), pp. 1095-1101.

Koarashi, J., Akiyama, K., Asano, T., Kobayashi, H. Chemical composition of 14C in airborne release from the Tokai reprocessing plant, Japan (2005) Radiation Protection Dosimetry, 114 (4), pp. 551-555.

Stenström, K., Erlandsson, B., Hellborg, R., Wiebert, A., Skog, S., Vesanen, R., Alpsten, M., Bjurman, B. A one-year study of the total air-borne14C effluents from two Swedish light-water reactors, one

boiling water- and one pressurized water reactor (1995) Journal of Radioanalytical and Nuclear Chemistry Articles, 198 (1), pp. 203-213.

Uchrin, G., Hertelendi, E., Volent, G., Slavik, O., Morávek, J., Kobal, I., Vokal, B. 14C measurements at PWR-type nuclear power plants in three middle European countries(1998) Radiocarbon, 40 (1), pp. 439-446.

We added as references Uchrin et al., 1998 and Povinec et at., 2009.

Pg 4 line 5-8. Please include references to back the statement that BWR reactors mostly emit 14CO2 whereas others emit 14CH4.

We added the original reference from Kunz, 1985

Pg 8 line 21-28. Are there previous studies that examined the performance of HySPLIT with met data at different resolutions? What did they conclude?

There have been earlier studies using HYSPLIT with differently resolved meteorological data, such as the one cited (Su et al., 2015, Science of the Total Environment 506-507, 527-537) however, their findings were not directly applicable to our problem.

Pg 11 line 5 and throughout. Through most of the paper, the nuclear facilities are identified by their names – "Phillipsburg", etc. Here they are identified by the 3 letter codes, which are particularly confusing since KPP is not obviously the same place as Phillipsburg. Choose either the names or 3 letter codes and stick with them throughout the text.

We have removed the 3 letter codes in the text and use now only real names of the facilities

Pg 14 lines 1-10. I agree that the detailed emissions and LaGrangian model used in this paper give more detail (and more variability) than Graven and Gruber showed in their earlier paper. Yet a little more nuance in this paragraph would be helpful. In cases where nuclear facilities are nearby and have a strong influence, the detailed studies such as this one will be necessary. But for continental-scale studies looking at monthly or annual resolution, the gridded datasets provided by Graven and Gruber will likely be sufficient – and in many cases, it may be difficult to get more detailed information, so the Graven and Gruber dataset may still be the best choice.

We do not fully agree to the reviewer: We rather think that a coarse-resolution Eulerian model, similar to that used by Graven and Gruber, is not able to provide reliable results, neither in the near (10s of km) nor in the far field (few 100s of km), simply because - with a spatial resolution of 1.8° x 1.8° -  it is principally not suited to simulate properly dispersion from a point source. It may be valuable to estimate the (very diluted) signal at the scale of 1000 km or so. Therefore, we think that for a reliable correction for nearby NPP contamination either a simple ("high-resolution") Gaussian plume approach (up to 10 km) or a high-resolution Lagrangian model is needed, preferably with higher resolution wind fields than used in the current study.

---

## Author Comment (AC2) · 30 Apr 2018

Replies to anonymous Referee #2

We wish to thank Referee #2 for her/his comments and suggestions for changes; we have revised the manuscript as follows (our answers are given in blue in the text below)

Kuderer et al. present an analysis of nuclear power plant influences on radiocarbon measurements in CO2 at Heidelberg using emissions data and the Hysplit model at three resolutions. Their main conclusions are that the nuclear correction decreased after the shutdown of Philippsburg BWR, the corrections they estimate are sensitive to model resolution, and nuclear corrections require careful consideration. The authors' work is useful and important to the community. However, some revisions are needed to clarify the details of their study and to expand the conclusions drawn from their results. The methods for model simulations are not very clear and there appear to be several different simulations used that are rather hard to follow.

A table describing the different simulations run for each nuclear site would be helpful.

We added a table (Tab. 2) with the respective information

Details about how the Hysplit runs were conducted, such as the number of particles and release times should be added.

The authors should also clarify that Hysplit was run in forward mode from the locations of the nuclear sites rather in backward mode from the observation site in Heidelberg. There appears to be some details described in the results section 3.2 that would fit better in the methods section.

The requested information is now added in the methods section 2.4 with some technical information moved here from the results section 3.2

The authors report in the abstract that "The mean correction for the period from 1986- 2014, if based on the 0.5∘ x 0.5∘ wind field, which we assume as the most accurate, is 2.3 ‰''. However, it appears high resolution 0.5∘ winds were only used in simulations for 2009 and 2011 – 2014, so it is not correct to say the 1986-2014 correction is from the 0.5∘ x 0.5∘ wind field. The other years were estimated from the coarser 2.5∘ resolution simulations with a correction factor based on comparisons for the years where 2.5∘ and 0.5∘ simulations were run for two reactors.

This is correct, and we changed the abstract as follows:

The finally applied mean $\Delta^{14}CO_2$ correction for the period from 1986-2014 is 2.3 ‰ with a standard deviation of 2.1 ‰ and maximum values up to 15.2 ‰. These results are based on the 0.5° x 0.5° wind field simulations in years when these fields were available (2009, 2011-2014) and, for the other years, they are based on 2.5° x 2.5° wind field simulations, corrected with a factor of 0.43.

Since Fig 4 shows the difference between simulated corrections at different resolution for individual samples is sometimes very small and sometimes very large (even with fixed emissions), is it valid to apply a mean correction to the data before 2009? Particularly if a main argument the authors are making is that the correction is highly variable in time? The authors argue that, since the correction is highly variable in time, monthly emissions data must be used and average emissions cannot be used, but then seem to contradict themselves by saying an average correction can be applied to account for model resolution, when actually this can be highly variable as well.

The referee is absolutely correct, however, we simply do not see an alternative possibility to correct for that obvious bias when using the coarse resolution wind field. We added in section 3.2 the expression … factor of 0.43, "as an attempt" to account for …

Another point is that 0.5∘ is still rather coarse compared to some regional modelling currently being done at 0.1∘ or finer resolution.

Yes, correct, however, higher resolution wind fields than those used here have not been available to us. We would be happy to work together with modelers in the future to refine our corrections …

Why do the authors use fixed emissions in the simulations shown in Fig 4?

We wanted to investigate the sole influence of the different wind fields on the correction (displayed in the more intuitive ‰ units). Further, including variable emissions would have led to a bias for high emission months.

If the authors have simulations with both fixed and monthly-varying emissions, can they include a comparison of these two simulations to quantify variability due to varying emissions vs variability due to varying transport? This comparison would be very useful.

All simulations are done with fixed emissions, which are later scaled. Sole emission variability is displayed in Fig. 2b, sole transport variability in Fig. 4a, and total scaled variability in Fig. 5b.

The authors note the previous estimate of the average nuclear correction by Levin using the plume model is higher than their estimate. Although the plume model is simpler, it might be considered to be at finer resolution than 0.5∘, and therefore a better estimate.

Levin et al. (2003) simply used one single dilution factor taken from the Turner (1970) workbook, and the mean wind statistics at the Philippsburg facility. The estimated factor, especially at the relatively large distance of 25 km from the emission point, may easily be wrong by a factor of two. (See also last comment to Referee #1.)

Can the authors make any inference on the detectability of the Philippsburg shutdown based on the Heidelberg Delta14CO2 data?

We have looked at that, however, as the average NPP contamination is of the same size as the individual measurement uncertainty, and as the variable fossil fuel signal is generally one order of magnitude larger, this is difficult or impossible.

The authors should discuss the impact the Philippsburg shutdown would have on the inferred fossil fuel CO2 at Heidelberg, if the change in the nuclear correction after 2011 was not accounted for. How does the change in the nuclear correction compare to the average fossil fuel signal in Delta14CO2 at Heidelberg?

(2.3 − 0.4 ‰)/(1.8‰/ppm) ≈ 1 ppm, this corresponds to ca. 10% of the total fossil fuel signal.

We added a respective remark at the end of the conclusions.

Section 3.3 Uncertainty in estimated nuclear correction – this needs more detail and seems rather too qualitative. The authors do not seem to include model transport uncertainty also for the high-resolution case.

Yes, but we are, unfortunately, not able to add more quantitative uncertainty estimates here, particularly not of model transport errors.

Do the authors have an estimate for the magnitude of sub-monthly variation in emissions?

No, the measurements in the exhaust air of Philippsburg reactors are integrated monthly values, and we are not aware of high-resolution exhaust data of $^{14}CO_2$. During revisions or fuel element change, there may occur short-term activity maxima, smeared out in the monthly means.

Do the authors think that monthly resolution in emissions data is sufficient, in general? Would this depend on the sampling integration time?

Emission data should ideally be as highly resolved as temporal changes of the meteorology occur, this may be hours or days. However, we think that for $^{14}C$ emissions we will probably not get higher than monthly or, at best, weekly resolved data, because $^{14}C$ measurements are (currently) not made continuously in situ, but rather on grab samples. We do not think that the required resolution would depend on the sample integration time, because the contamination at the measurement site varies most with meteorological conditions.

Could Fig 2b show the time series of emissions rather than yearly boxplots? It would be interesting to know if there is any pattern to the emissions over the year – for example, are emissions typically higher in summer potentially related to more maintenance undertaken in summer?

There is no apparent seasonality in the monthly emission data and no clear relationship between emissions and maintenance intervals. We added a respective sentence in the revised manuscript.

The authors should consider the Cattenom reactor in France, to the west and upwind of Heidelberg.

The estimated $^{14}CO_2$ emissions of the 4 PWR blocks of Cattenom (ca. 5.4 GWe) is about 0.2 TBq/year, i.e. half of the emissions from Philippsburg I. These reactors are located around 170 km west of Heidelberg. Therefore, we do expect at least one order of magnitude lower contamination from these facilities compared to those estimated for Philippsburg I.

Shouldn't Eq. 3 have a factor of 1000 for per mil units? What is used for XCO2 in this C3 calculation?

We have added the factor 1000. We used individually measured two-weekly mean $CO_2$ mole fractions.

First sentence in abstract – last word "fluxes" should be deleted. Radiocarbon measurements quantify fossil fuel derived CO2, but not fluxes.

We have changed "fluxes" into "component"

Also here the Delta notation is used without describing it. The phrase "14CO2 signal" is unclear – do you mean nuclear Delta14CO2 signal? Why are Delta14CO2 and 14CO2 both used in the abstract?

Isn't the Heidelberg site monitoring Delta14CO2 rather than 14CO2?

We think we can use in qualitative cases $\Delta^{14}CO_2$ and $^{14}CO_2$ more or less synonymously; however, we have made it clearer now that we give all numbers on the $\Delta$-scale

Last sentence should be revised to "After operations at the Philippsburg boiling water reactor ceased in 2011, the" . . .

We changed this sentence accordingly

P2, L22: Delete "well".

Not sure why - the signals can really be large close to the facilities, i.e. a few km away !

L24 Comment about "normally quickly disperse" needs reference or should be deleted.

References were added.

P10, L21-24 – Please show some quantitative evidence from the simulations to support these statements.

Statements: "Therefore, and considering the high month-to-month variability of emissions (Fig. 2, lower panel), it is important to use monthly resolved emission data to estimate the $\Delta 14C_{nuclear}$ signals originating from KKP I & II. The other four nuclear installations are secondary contributors permitting the use of annual average $14CO_2$ emission rates in absence of higher temporally resolved emission data."

As the variability in model transport and NPP emissions are independent, we have to assume that both variabilities contribute to the total variability.

Figure 4b directly shows the low contamination from Neckarwestheim, which is only about 20% of that from Philippsburg (due to the ca. 25% lower emission rate and the larger distance of this NPP from Heidelberg).